

# Model simulations with COSMO-SPECS: Impact of heterogeneous freezing modes and ice nucleating particle types on ice formation and precipitation in a deep convective cloud.

Karoline Diehl[1] and Verena Grützun[2]

[1]Institute of Atmospheric Physics, University of Mainz, Germany
[2]Institute of Meteorology, Hamburg, Germany

*Correspondence to*: Karoline Diehl (kdiehl@uni-mainz.de)

**Abstract.** In deep convective clouds, heavy rain is often formed involving the ice phase as essential process. Simulations were performed using the 3D cloud resolving model COSMO-SPECS with detailed spectral microphysics including parametrizations of homogeneous and three heterogeneous freezing modes. The initial conditions were selected to result in a deep convective cloud reaching 14 km altitude with strong updrafts up to 40 m/s. In such altitudes with corresponding temperatures below -40°C the major fraction of liquid drops freezes homogeneously. The goal of the present model

simulations was to investigate how additional heterogeneous freezing will affect ice formation and precipitation although its contribution to total ice formation may be rather low. In such a situation small perturbations which do not show significant effects at first sight may trigger cloud microphysical responses. Effects of the following small perturbations were studied: (1) additional ice formation via immersion, contact, and deposition modes in comparison to sole homogeneous freezing, (2) contact and deposition freezing in comparison to immersion freezing, (3) small fractions of biological ice nucleating

particles (INP) in comparison to higher fractions of mineral dust INP. The results indicate that the modification of precipitation proceeds via the formation of larger ice particles which may be supported by direct freezing of larger drops, the growth of pristine ice particles by riming, and by nucleation of larger drops by collisions with pristine ice particles. In comparison to the reference case homogeneous freezing such small perturbations may affect an enhancement of total precipitation but mostly the effects are limited to modifications of the temporal development of precipitation, i.e. a gradual

increase already at early cloud stages instead a strong increase at later cloud stages. These effects are coupled with changes in the local distribution of precipitation, i.e. approximately 50% more precipitation in the cloud center. The modifications depend on the active freezing modes, the fractions of active INP, and the composition of the internal mixtures in the drops.



# 1 Introduction

Deep convective clouds may cover a wide temperature range from +20°C at ground level down to -40°C at altitudes of 14 km. The high vertical updraft in these clouds transports moist air to high levels where most of the water vapor is condensed leading to total water contents as high as 10 g/m$^3$ (Wu et al., 2000). During early cloud stages, the condensed water is present in form of liquid droplets but after passing the zero degree level mixed phase conditions are established where ice particles and supercooled liquid drops are present simultaneously (e.g., Rosenfeld and Woodley, 2000). Heterogeneous freezing may be active at temperatures below -2°C in dependence of freezing mode and involved ice nucleating particles (INP) (Hoose and Möhler, 2012). When reaching altitudes with temperatures below -37°C most of the liquid water is changed into ice by homogeneous freezing (e.g., Pruppacher and Klett, 2010). The ice mass increases further by growth processes such as the deposition of water vapor and supercooled droplets (i.e. riming) on ice particles and by the nucleation of supercooled drops via collisions with small ice particles. These processes lead to the formation of increasingly larger ice particles which eventually transfer the melting layer and result in heavy precipitation.

The distribution of liquid and ice water mass is dependent on factors such as altitude, temperature, and in particular aerosol and ice nucleating particle concentrations and composition as well as on the active freezing modes (e.g., Khain et al., 2005; Leroy et al., 2006; Tao et al., 2007; Fan et al., 2013, Hiron et al., 2015). In deep convective clouds, a large fraction of ice is formed homogeneously; however, heterogeneous freezing already at lower altitudes may have important effects on ice formation and, thus, precipitation (e.g., Gilmore et al., 2004; van den Heever et al., 2006; Ekman et al., 2007; Phillips et al., 2007; Lee et al., 2009).

The present model simulations follow the question how additional heterogeneous freezing will affect ice formation and precipitation although its contribution to total ice formation may be rather low. This situation may create so-called "small trigger effects", i.e. small perturbations which do not show significant effects on the first sight may trigger cloud microphysical responses. Small perturbations may also play a role within heterogeneous freezing processes themselves. Immersion freezing is assumed to represent the most important heterogeneous freezing process (e.g. Phillips et al., 2007). However, even small additional contributions from contact and deposition freezing may alter eventually precipitation. Another situation with small perturbations is the composition of ice nucleating particles (INP). It was shown that certain aerosol types significantly modify cloud microphysics (e.g., Lohmann and Diehl, 2006; Phillips et al., 2008; Lee et al., 2009; DeMott et al., 2015; Hande et al., 2015). The most important INP types are mineral dust and biological particles but the latter are present in the atmosphere in much lower amounts than mineral dust particles (e.g., Phillips et al., 2009; Paukert and Hoose, 2014). Thus, the low fractions of biological particles (BAP) may trigger significant effects.

Model simulations dealing with these issues were performed with the state-of-the-art model system COSMO-SPECS, a 3D cloud model developed by Grützun et al. (2008). A follow-up version which was numerically more effective was provided by Lieber et al. (2012). COSMO-SPECS is well suited for the envisioned investigations as it provides a link between aerosol particles, cloud properties, and precipitation. It contains a detailed description of the cloud microphysical processes,



achieved by a spectral bin-microphysics that explicitly solves the microphysical equations. The last versions of COSMO-SPECS included parameterizations of immersion and contact freezing for several particle types such as mineral dust, soot, and biological particles from Diehl et al. (2006). Recently, in Diehl and Mitra (2015) the parameterizations of ice forming processes were extended and improved. They include now deposition nucleation and homogeneous freezing as new ice

forming processes as well as advanced descriptions of immersion and contact freezing. For the present investigations, this new version of the microphysics was implemented in COSMO-SPECS.

The model simulations presented here are part of the German Science foundation (DFG) research group INUIT (**I**ce **N**uclei Research **U**n**IT**) which was established in 2012 to study heterogeneous ice formation in laboratory, field, and model studies. As an outcome of the experiments, joint parameterizations were derived to be fed into cloud models to simulate mixed-phase

cloud microphysics. For more details see the INUIT website: www.ice-nuclei.de.

## 2 Model description

### 2.1 Previous version of COSMO-SPECS

The COSMO model (Consortium for Small-scale Modeling; Steppeler et al., 2003; Baldauf et al., 2011) is the regional part of the operational weather forecast system of the DWD. It is based on the primitive hydro-thermodynamic equations

describing compressible non-hydrostatic flow in a moist atmosphere (www.cosmo-model.org).

The original COSMO model works with a Kessler-type cloud microphysics bulk scheme including various states of water (cloud and rain water, several forms of ice). Grützun et al. (2008) completely replaced the bulk scheme by a spectral bin microphysics as described in Simmel and Wurzler (2006) and Diehl et al. (2006). The time integration of the coupling scheme between the COSMO model and the bin microphysics is performed with two different time steps because the

microphysics operates on much smaller time scales than the concurrent dynamical processes. Within the COSMO model the horizontal and vertical winds as well as temperature and pressure are transported within a time step of 10 to 100 s leading to dynamically updated values. These are used for the microphysical loop which consists of time steps of 1 s or smaller where changes in the hydrometeor spectra due to the included microphysical processes are calculated (Grützun et al., 2008).

The spectral cloud microphysics describes all microphysical processes during the development of clouds and the subsequent

initiation of precipitation. The entrainment of aerosol particles, drops, ice particles, temperature, and humidity is embedded (Simmel et al., 2005). A fixed bin structure is used where in a first spectrum wetted aerosol particles and liquid drops are combined. An initially dry aerosol particle number size distribution is defined where the particles are internally mixed with a soluble fraction ε. By condensation, the particles grow into the droplet part of the spectrum. The size spectra are allowed to evolve freely, they are not constrained by underlying distribution functions. Thus, the particles and drops move in this

spectrum by processes such as growth by water vapor deposition, shrinking by evaporation, collision and coalescence of drops, and impaction scavenging of particles. Ice particles are formed from supercooled liquid drops via immersion and contact freezing described in parameterized form (Diehl and Wurzler, 2004; Diehl et al., 2006). Condensation freezing is





included implicitly in immersion freezing: drops which are nucleated by aerosol particles entrained above the freezing level could freeze immediately by immersion freezing.

After freezing, the drops are removed from the first liquid spectrum and shifted into a second spectrum which is used for mixed-phase particles. These consist of an ice core and a liquid shell; the liquid water mass may be zero to describe

completely frozen particles. In the mixed-phase spectrum (with the same bins as the liquid spectrum) particles move by processes such as growth by water vapor deposition and by riming (i.e., collision with smaller supercooled droplets), collision and sticking of ice particles, ice nucleation of supercooled drops by collision with smaller ice particles, sublimation, and melting. This latter process is modeled by the possible existence of a liquid water shell. In this study, both spectra are divided into 66 categories, starting with 0.002 µm in diameter, with a mass doubling in every category.

Collision processes are described by the linear discrete method (Simmel et al., 2002) including the collision kernel of Kerkweg et al. (2003). By using the corresponding densities and terminal velocities, the collision kernel is appropriate for all collision processes between aerosol particles, drops, and ice particles such as collision/coalescence of drops, impaction scavenging of particles by drops, contact freezing of supercooled drops after collisions with particles, riming of ice particles by collisions with supercooled liquid droplets, nucleation of supercooled drops by collisions with small ice particles, and

sticking of ice particles after collision.

## 2.2 Improvements of ice parameterizations in COSMO-SPECS

### 2.2.1 Homogeneous freezing

In the new version of COSMO-SPECS, drops may freeze homogeneously at temperatures below -37°C. This is determined by the soluble particle fraction dissolved in the drops together with the drop volume. A lower solute content and a larger

drop volume affect higher freezing temperatures (Koop et al., 2000; Duft and Leisner, 2004). Homogeneous freezing is described in the model according to the approach of Koop et al. (2000). With their parameterization of the water activity criterion the freezing temperatures of solution drops in dependence on their molality are calculated (Diehl and Wurzler, 2004).

### 2.2.2 Immersion freezing

The parameterization of immersion freezing in Diehl and Mitra (2015) is an updated version which is related to the insoluble particle mass in drops. It is based on laboratory data of $n_m(T)$, the number of active sites per unit mass at temperature $T$. As shown in Fig. 1a, $n_m$ exponentially increases with temperature $T$ as described by

$$n_m = \exp\left(a_{imm} + b_{imm}T_s\right) \tag{1}$$

with $n_m$ in g$^{-1}$, $a_{imm}$ and $b_{imm}$ particle-related constants, $T_s = T_0 - T$, $T_0 = 0$°C, with $T$ in °C. The constants are given in Diehl

and Mitra (2015) together with two more parameters: $T_{ini}$ represents the onset of immersion freezing during experiments, $T_{lim}$ represents the temperature where $n_m$ reaches a plateau value (Wex et al., 2015) see Fig. 1a. A new particle type is





included here based on measurements of cellulose (Hiranuma et al., 2015) which represent a macro-tracer for plant debris (with the corresponding constants $a_{imm}$=7.86464, $b_{imm}$=0.560, $T_{ini}$ =-10°C, $T_{lim}$ = -36°C). The freezing rate of drops containing insoluble ice-nucleating material is given by (Diehl and Mitra, 2015):

$$\frac{dN_f}{dt} = N_{liq} \frac{1 - \exp\left(-K(T)\, m_{pid}\, F_{INP}\, dT\right)}{dt} \qquad (2)$$

with $N_f$ the number of frozen drops, $N_{liq}$ the number of liquid drops, $t$ the time, $m_{pid}$ the insoluble particle mass immersed in the drops, $F_{INP}$ the mass fraction accounting for possible numbers of ice-active sites. $K(T)$ stands for the cumulative nucleus spectrum per unit mass per unit temperature which is related to $n_m$ by:

$$K(T) = \frac{dn_m(T)}{dT} = b_{imm}\ \exp\left(a_{imm} + b_{imm}\, T_s\right) \qquad (3)$$

As in the previous version, condensation freezing is included implicitly as it is also initiated by an INP immersed in a supercooled drop. The difference is a temporal separation (Cziczo and Froyd, 2014): if directly after drop activation the insoluble particle mass is sufficient for drop freezing at the actual temperature, the drop freezes immediately. If the ambient temperature is too high, it may freeze later at lower temperatures.

### 2.2.3 Contact freezing and deposition nucleation

The description of contact freezing was modified by Diehl and Mitra (2015) in the way that it is also particle size-resolved. A particle-type dependent parameterization of deposition nucleation was newly added (Diehl and Mitra, 2015). Because of entrainment inactivated interstitial particles are always present during the simulations with COSMO-SPECS and may serve as contact and deposition ice nucleating particles. If during the model simulations particles collide with supercooled drops the number of frozen drops formed by contact freezing is calculated according to:

$$N_f = F_{INP}\, N_{liq} \left(a_{con}\, T + b_{con}\right) \qquad (4)$$

with $N_f$ the number of frozen drops, $N_{liq}$ the number of liquid drops, $T$ the temperature, $a_{con}$ and $b_{con}$ particle and size-related constants (see Diehl and Mitra, 2015), $F_{INP}$ the ice-active fraction of the aerosol particles. Figure 1b shows the freezing probability for different particle types and sizes. New particle types are plant debris and pollen based on laboratory measurements of Hoffmann (2015) and Hiranuma et al. (2015); their constants are listed in Table 1.

Interstitial particles may also serve for deposition nucleation. According to experimental findings the number of activated particles increases exponentially with ice supersaturation which is shown in Fig. 1c and calculated by (Diehl and Mitra, 2015):

$$N_{act} = F_{INP}\, N_{total} \exp\left(a_{dep} + b_{dep}\, s_{ice}\right) \qquad (5)$$

with $N_{act}$ the number of activated particles, $N_{total}$ the total particle number, $F_{INP}$ the fraction of ice-active particles, $s_{ice}$ the ice supersaturation given in %, $a_{dep}$ and $b_{dep}$ particle-related constants. The constants are given in Diehl and Mitra (2015) together with two more parameters: $T_{ini}$ and $s_{ini}$ represent initial values of temperature and ice supersaturation for the onset of





deposition freezing during experiments. The activated particles are shifted to the mixed-phase spectrum and grow further by water vapor deposition and riming and they may initiate freezing of supercooled drops by collision.

For potential contact and deposition ice nucleating particles minimum sizes are defined: for mineral dust particles 0.1 µm, for bacteria 0.3 µm, for plant debris 0.35 µm (Matthias-Maser and Jaenicke, 1995), and for pollen 0.4 µm. Complete pollen

grains are large particles of 10 µm at least (Straka, 1975); however, Steiner et al. (2015) indicated the existence of so-called sub-pollen particles due to a pollen grain rupture after wetting.

## 3 Model initial conditions and process studies

### 3.1 Convective cloud, vertical profiles, and particle number size distribution

With COSMO-SPECS idealized test cases were simulated. A heat bubble over a flat terrain was initialized by a temperature

disturbance of 1.5K which resulted in a deep convective cloud. The complete model domain covered $80 \times 80$ km$^2$ with a horizontal resolution of 1 km. The model top reached an altitude of 18 km with a vertical resolution between 100 m and 600 m (48 levels). The heat bubble was located in the domain center at 1.4 km height and had a horizontal extension of 20 km and a vertical extension of 1.4 km. The initial wind was set to zero. The vertical profiles of temperature and dew point from Weisman and Klemp (1982) which are consistent with real conditions in convective situations are shown in Fig. 2a.

As initial dry aerosol particles the number size distribution of Kreidenweis et al. (2003) was selected. As can be seen from Fig. 2b, it is a mono-modal lognormal size distribution with N = 566 cm$^{-3}$, d = 30 nm, and σ = 2 (with N the particle number, d the median diameter, and σ the standard deviation). After starting the simulation, the particle spectra evolved freely within the given size ranges. The soluble fraction ε of the aerosol particles was set to 0.5 which is a typical value of atmospheric particles (Busch et al., 2002). The ice nucleating particles are defined as part of the complete aerosol particle

spectrum. The insoluble fraction of the particles entering drops via nucleation or impaction scavenging accounts for immersion freezing. Interstitial aerosol particles may serve as contact and deposition ice nucleating particles. Because of the size conditions of the ice nucleating particles (see Sect. 2) one can note from Fig. 2b that the majority of the particles was not suited to initiate ice as it is also the case in the real atmosphere.

### 3.2 Freezing processes, INP types and fractions

Process studies were performed including various ice forming processes and, in case of heterogeneous freezing, different ice nucleating particle types in various fractions. First, a warm test case without freezing was simulated to characterize the behavior of the deep convective cloud. Afterwards, a case with homogeneous freezing only was performed which served as reference case. To study the characteristic impacts of the individual heterogeneous freezing processes, simulations without homogeneous freezing were performed although these do not represent realistic cases. The next step was to couple one

heterogeneous freezing process with homogeneous freezing, afterwards two or all three heterogeneous freezing processes with homogeneous freezing. This kind of stepwise adding ice forming processes allows the study of the impact of small



perturbations of less active freezing processes such as contact and deposition freezing. More small perturbations are low numbers of ice nucleating particles and, in particular, biological particles. Therefore, for each type of simulation, the following parameters were varied:

- ice nucleating particle type – biological particles and mineral dust,
- $F_{INP}$, the ice-active fraction of aerosol particles.

As examples for the present paper only three types of mineral dust were selected, feldspar, kaolinite, and Saharan dust. Feldspar represents a very effective INP type which is contained in desert dusts and also in illite samples. Therefore, by scaling down it is representative for dust samples in dependence on their composition (Atkinson et al., 2013). E.g., African and Asian dust contains around 24% feldspar, Arizona test dust (ATD) approximately 20%, illite NX 14%. Kaolinite samples may also include up to 10% feldspar, but CMS kaolinite which was used for the experiments which served as base for the present parameterization does not show detectable amounts (Murray et al., 2011). Therefore, it shows a significantly lower efficiency than feldspar in immersion and contact freezing (see Figs. 1a, 1b) and was used in these modes together with feldspar to indicate the lowest and highest effects. For deposition nucleation no parameterization of kaolinite is available, therefore, less efficient INP are represented by Saharan dust (see Fig. 1c). Biological particles are represented by bacteria, plant debris, and pollen.

To reflect atmospheric conditions, the ice active-fractions of mineral dust were larger than the ones for biological particles. $F_{INP}$ values for mineral dust ranged from 0.1% to 10%, for biological particles from 0.001% to 0.01%. These values were used already in Diehl and Mitra (2015) according to results from field measurements of cloud droplet residuals and background aerosols (e.g., Bauer et al., 2002; Twohy and Anderson, 2008; Kamphus et al., 2010; Hiranuma et al., 2013, Schmidt et al., 2017). In both cases the highest values are slightly overestimated while the other values represent realistic situations.

## 4 Results and discussion

### 4.1 Warm test case

As first test case, a warm case was performed where all freezing processes were switched off. This study demonstrated the formation of a deep convective cloud where the cloud top reached 14 km altitude with temperatures of -50°C. Figure 3, upper and middle panel, shows a vertical cut through the cloud center and illustrate the development of the liquid water content and the vertical velocity with time. The dotted lines give the temperature levels; note that they are lifted up inside the cloud because of the initial temperature disturbance and convective transport. After 15 min the cloud top passed the 0°C-level in 4 km altitude, after 30 min the cloud reached its maximal top height of 14 km. Precipitation set in after 45 min, after 60 min the cloud started to dissipate as the cloud top height was decreasing. Correspondingly the strongest vertical updraft in the cloud was noted after 30 min with vertical velocities up to 40 m/s in the cloud center which agrees with values found by Weisman and Klemp (1982). The complete aerosol and drop size spectra in the center cell of the cloud with the drop radius on the x-axes is given in the lower panel of Fig. 3. The gap in the spectra at 1 μm radius indicates the distinction between





aerosol particles and drops. Note that the majority of the drops stayed smaller than 100 μm while parts of them grew to larger sizes in the millimeter range by collision and coalescence.

## 4.2 Single homogeneous and heterogeneous freezing

In a deep convective cloud as presented in Sect. 4.1 the major fraction of liquid water freezes homogeneously (Phillips et al.,
2007). In the present study for a reference case only homogeneous freezing was switched on, occurring at temperatures below -37°C, i.e. at altitudes above 9 km. Afterwards, simulations were performed where homogeneous freezing was switched off and only one single heterogeneous freezing process was switched on. To decide which cases to select for demonstrating possible effects of small perturbations the ice water fractions in the resulted clouds were determined for a number of cases.

### 4.2.1 Ice water fractions

Following the definition of Korolev et al. (2003) the ice water fraction decides whether a liquid or a mixed-phase cloud has been formed. It is calculated from the integrated ice water content $IWC$ and the integrated liquid water content $LWC$ by:

$$IWF = \frac{IWC}{LWC + IWC} \tag{6}$$

Values below 0.1 define a liquid cloud, values above 0.1 a mixed-phase cloud (Korolev et al., 2003). The resulting types of
clouds are listed in Table 2. Homogeneous freezing resulted in a mixed-phase cloud as well as immersion freezing with mineral dust fractions as low as 0.1% and biological fractions as low as 0.01% except pollen. With 0.001% biological fractions bacteria did still form a mixed phase cloud but not plant debris and pollen. In contact and deposition modes, mixed-phase clouds were found only with 10 and 1% feldspar and 10% kaolinite/Saharan dust; in all other cases liquid clouds resulted, even with somewhat higher biological fractions of 0.01%. Therefore, the biological particles were not included in
simulations with contact and deposition freezing. The results in Table 2 indicate that cases representing small perturbations were those with 0.001% biological material in the immersion mode and those with 1% mineral dust in contact and deposition modes.

In the following sections, some example results from these simulations are presented: for the reference case homogeneous freezing, for immersion freezing with 1% feldspar, and for contact and deposition freezing with 10% feldspar. Figures 4 to 7
show the corresponding results of ice formation. The figures in the upper panels show the ice water contents in g/kg after 30, 45, and 60 min in a vertical cut through the cloud center. The figures in the middle panels give the ice particle numbers per $m^3$ in a vertical cut through the cloud center after 30 and 60 min, the figures in the lower panels show the ice particle size spectra in the center cell of the cloud with number concentrations per $m^3$ after 30 and 60 min. The pictures in the left columns of the middle and lower panels indicate results from primary freezing only, i.e. from direct drop freezing
(homogeneous, immersion, and contact freezing) or direct particle activation (deposition nucleation). The other pictures in



the middle and lower panels give results from complete ice formation including growth by water vapor deposition, riming, collision and sticking of ice particles, and ice nucleation of drops by small ice particles (see Sect. 2.1).

### 4.2.2 Ice water contents

The maximum ice water contents (IWC) reached 10 g/kg in all cases (Figs. 4 to 7, upper panels). Thus, with contact and deposition freezing around 10 times more ice nucleating material as with immersion freezing was required to affect similar IWC. The homogeneous case, however, showed the largest regions with 10 g/kg ice water content in altitudes above 10 km even after 45 min. Later, the region with more than 0.1g/kg IWC was still enlarged.

In cases of contact and deposition freezing, after 30 min the cloud regions with more than 0.1 g/kg IWC were smaller in comparison to homogeneous and immersion freezing. In particular there was a gap in the center of the cloud: because of high relative humidities in this cloud region less interstitial (i.e. inactivated) aerosol particles were present which could have served as INP. After 45 min, melting ice particles arrived near ground level in case of contact and deposition freezing and also, but less, in case of homogeneous freezing, but not with immersion freezing. In the latter case less ice particles were removed from the cloud, even at later cloud stages.

### 4.2.3 Ice particle numbers

Figures 4 to 7, middle and lower panels, left columns, indicate the altitude where primary ice formation proceeded. The maximum in the homogeneous case was between 10 and 12 km altitude (-40 to -45°C), in the deposition case above 13 km due to high ice supersaturations and corresponding low temperatures (< -45°C) which were present near cloud top. However, some particles were nucleated also at lower heights at the cloud edges. Immersion freezing was active in a wider range between 8 and 11 km with temperatures from -20 to -40°C according to the insoluble particle mass in the drops. Contact freezing was dominant in lower altitudes from 6 to 9 km with temperatures between -10 and -25°C due to the effects of large particles colliding with drops. Note from Fig. 6, lower panel, the second maxima in the ice particle spectra at 12 km altitude which reflect the effects of small particles at low temperatures.

In homogeneous, immersion, and contact modes most primary frozen drops had radii around 40 μm; however, the complete ice particle spectra were rather different. Homogeneously frozen drops started with 1 μm radii while the major fraction had radii between 10 and 100 μm. With immersion freezing the size spectrum was much broader ranging from 40 μm to 1 mm radii; the smallest frozen drops had 10 μm because they had to sample sufficient ice nucleating material. The major fraction of drops frozen by contact freezing had radii from 20 to 80 μm also starting with 10 μm. For smaller drops, the collision efficiency with ice nucleating particles is very low (Diehl et al., 2006). In contrast, primary formed ice particles from deposition nucleation had radii around 0.1 to 0.2 μm only due to the sizes of involved aerosol particles.

For the homogeneous and immersion case, the ice water contents as well as the ice particle numbers decreased between 30 and 60 min (Figs. 4 and 5, upper and middle panels) while for the contact and deposition cases the IWC decreased but the ice particle numbers did not. This indicates that primary ice formation still continued at cloud stages after 60 min with contact




and deposition freezing. Homogeneous and immersion freezing occurred mainly in altitudes above 9 km where the numbers of available supercooled drops were reduced after 60 min (compare the warm test case in Sect. 4.1). Inactivated particles required for contact and deposition freezing were always present because of entrainment. Furthermore, for contact freezing taking place in lower altitudes still supercooled drops were available.

With homogeneous and immersion freezing, high numbers of ice particles were formed (Figs. 4 to 7, middle panels): up to 1 $\times 10^6$ ice particles per m$^3$ in the homogeneous case, up to $1 \times 10^4$ ice particles per m$^3$ in the immersion mode. These numbers are in the range of those observed in atmospheric convective clouds (e.g., Frey et al., 2014). Similar numbers as with immersion freezing were reached with deposition nucleation but only in a limited area of the cloud where in contrast the IWC was low. This indicates that these ice particles were very small and contributed very little to the IWC. With contact

freezing, maximum only 300 ice particles per m$^3$ were formed. Consider here that the INP fractions were ten times higher in contact and deposition cases than in the immersion case.

After 60 min, in the homogeneous case still small ice particles were present in high altitudes which were grown from the very small ones; larger ice particles moved downwards (Fig. 4, lower panel). In contrast, in the immersion case most of the smaller ice particles grew to larger sizes and moved downwards (Fig. 5, lower panel). With contact freezing, still newly

formed smaller ice particles were present at lower altitudes after 60 min. In the deposition mode, an important process was the nucleation of larger drops by collisions with the pristine ice particles. These effects are visible in particular in Fig. 6, lower panel, after 30 min where the ice particle spectrum was separated into two regions. All ice particles larger than 100 μm were the result from secondary ice formation, i.e. the collision of pristine ice particles with supercooled drops. In contrast, with homogeneous, immersion, and contact freezing, such large ice particles could be the result of primary ice formation

(see Figs. 4 to 6, lower panels) but this was significant only for immersion freezing. With contact freezing a higher fraction of large ice particles was probably also the result of secondary ice formation considering the partially separated ice particle spectrum.

These results from single homogeneous and heterogeneous freezing indicate that there is probably no competition between the different freezing processes because they occur at different altitudes and regions in the cloud. As the primary effects have

significantly different magnitudes one may assume that they do not affect each other, e.g., immersion freezing is not restricted by simultaneous contact freezing and conversely. Because of the fast updraft in the cloud the drop numbers in higher altitudes are hardly reduced by the small effects of contact freezing occurring in lower altitudes.

### 4.3 Coupled homogeneous and heterogeneous freezing with effects on precipitation

### 4.3.1 Single heterogeneous modes combined with homogeneous freezing

As a first step model simulations were performed with simultaneous homogeneous freezing and one heterogeneous mode, i.e. homogenous freezing was always switched on plus one heterogeneous mode. The total precipitation after 180 min modeling time was determined and compared to the value from the reference case with sole homogeneous freezing. Table 3



shows results for feldspar, kaolinite, Saharan dust, and, in the immersion mode, additionally for bacteria, plant debris and pollen. The results summarized in Table 3 indicate that in most cases the total precipitation amount was similar to homogeneous freezing while there were some cases with more than 20% deviations in both directions. In particular, enhanced precipitation after 180 min was found in the immersion mode for plant debris and pollen and in the deposition mode for 1% Saharan dust. These cases represent situations where small perturbations – here: small fractions of biological INP or little ice forming effects from deposition nucleation – trigger cloud microphysics in a way that eventually more precipitation is formed.

Figure 8 shows more precipitation details for the cases listed in Table 3. In Fig. 8a the development of total precipitation with time is given, Fig. 8b indicates the local distribution of precipitation on a longitudinal line through the model domain after 180 min. In all cases precipitation set in after 45 min (Fig. 8a). Deviations were already visible at that time but became more obvious with proceeding time. In some cases precipitation stayed constant during the next hour and increased at later times. This delayed increase of precipitation was noted for the reference case homogeneous freezing (black solid line), for the cases with contact freezing (purple lines), and for mineral dust cases with immersion freezing except 0.1% kaolinite (blue lines). In other cases, precipitation increased already at early cloud stages, in particular with biological particles in the immersion mode (green lines) and with Saharan dust in the deposition mode (yellow lines).

From Fig. 8b one notes that in the cloud center precipitation ranged from 65 mm (immersion with 1% kaolinite) to 160 mm (immersion with 0.001% plant debris) with 75 mm in the reference case. Higher precipitation in the cloud center was observed for the cases with at least 20% more total precipitation (see Table 3); however, it was found also for cases where total precipitation was not significantly enhanced but precipitation was increased during early cloud stages, i.e. 10% Saharan dust in the deposition mode, 0.001% bacteria and 0.1% kaolinite in the immersion mode.

To illustrate how ice formation influences the total condensed water in the deep convective cloud and, thus, precipitation, results from four example cases are shown. These are immersion with 1% feldspar (case 1), immersion with 0.001% plant debris (case 2), contact with 1% feldspar (case 3), and deposition with 1% Saharan dust (case 4). The amounts of total precipitation were $4.33 \times 10^9$ L, $8.81 \times 10^9$ L, $5.34 \times 10^9$ L, and $6.97 \times 10^9$ L, respectively. Figure 9 shows results from the cases 1 to 4 in each panel. The first three columns on the left give the ice water contents in g/kg after 45, 60, and 90 min in a vertical cut through the cloud center. The column on the right shows the ice particle numbers per m³ in a vertical cut through the cloud center after 60 min.

Differences between case 1 where precipitation was mostly delayed and the other cases are significant. In case 1 more ice was present in mid and high altitudes after 60 min but less ice near ground level after 45 and 60 min and in mid-levels after 90 min. Thus, only la few melting ice particles precipitated from the cloud. The ice particle spectrum after 60 min indicates that there was a high contribution from immersion freezing to ice particles smaller than 500 μm in mid-levels between 4 and 9 km height. The numbers of larger ice particles were reduced in comparison to the other cases; their formation was hindered by the competition of the many small ice particles.



In cases 2 to 4, the contributions from heterogeneous freezing to ice particles smaller than 500 μm in mid-levels between 4 and 9 km height were rather low in contrast to case 1. However, more ice was present near ground level after 45 and 60 min and below 8 km height after 90 min. This indicates the presence of larger ice particles which strongly contribute to the ice water content. After 60 min, the numbers of ice particles larger than 500 μm were highest in cases 2 and 4 where precipitation mostly increased already in early cloud stages. In case 3 with contact freezing less numbers of larger ice particles were formed. From single contact freezing (Sect. 4.2.1) it was found that primary contact freezing proceeded also at later cloud stages which hindered the growth of ice particles by competition.

The evaluation of the results indicates that the formation of precipitation-sized ice particles larger than 500 μm was essential to increase precipitation. The implications are advanced precipitation during early cloud stages, enhanced precipitation in the cloud center, and potentially higher total precipitation. Processes contributing mainly to the formation of larger ice particles are riming and the nucleation of supercooled drops by collision with pristine ice particles. The latter plays a major role with deposition freezing (see Sect. 4.2.1). In comparison to the reference case homogenous freezing, additional heterogeneous ice formation by direct drop freezing affected by high INP fractions may delay precipitation because growth of ice particles is hindered. On the other hand, small numbers of ice particles heterogeneously formed by low INP fractions trigger growth processes in the ice phase and, thus, may affect an increase of precipitation.

**Internally mixed INP in the immersion mode**

A number of cases were modeled with immersion freezing where the insoluble mass contained in the drops did not consist of pure materials but were internally mixed. These mixtures contained higher fractions of mineral dust and small fractions of biological particles as it reflects atmospheric conditions. Table 4 lists the compositions of seven cases together with the resulting amount of total precipitation; Fig. 10 shows the precipitation results.

In all mixed cases precipitation was lower than in the reference case homogenous freezing. However, as can be seen from Fig. 10, the temporal development and the local distribution of precipitation were modified by the particle composition. In cases 1 and 2 with 5% mineral dust fractions as well as in cases 3, 4, and 5 with 1% mineral dust fractions the development of precipitation was delayed below the reference line (homogeneous freezing) during early cloud stages (Fig. 10a). The lower the fraction of efficient dust INP the earlier followed an increase of precipitation above the reference line. This was distinctly visible in case 7 with the lowest dust fractions. Just as well, the 75 mm precipitation in the cloud center from homogeneous freezing were enhanced up to 100 to 120 mm in those cases. However, in comparison to the pure mineral dust cases no enhancement effects were resulted by additional biological INP fractions. This indicates that the major fraction of composed INP decides ice formation and hence, the development of precipitation.

### 4.3.2 Several heterogeneous modes combined with homogeneous freezing

Finally model simulations were performed where contact or/and deposition freezing were switched on additionally to homogeneous and immersion freezing. In a first series of model simulations the $F_{INP}$ values were 1% in all freezing modes.



For these cases Table 5 gives the resulting precipitation after 180 min together with results from Sect. 4.3.1 for sole immersion, contact, and deposition freezing. In a second series of model simulations mixed coupled cases were performed with low INP fractions in the immersion mode and higher INP fractions in contact and deposition modes. Here the immersion INP were internally mixed. Table 6 summarizes five selected cases where the INP fractions of approximately 0.1% in the immersion mode contained mainly feldspar (cases 1, 2, and 3) or kaolinite (cases 4 and 5). In cases 3 and 5, the mixed immersion INP contained additionally small fractions of biological particles. The INP fractions in contact and deposition modes were 1% mineral dust in cases 1, 2, and 4 and 0.1% biological particles in cases 3 and 5.

The results in Table 5 indicate that in none of the investigated cases additional contact or/and deposition freezing as small perturbations caused an increase of total precipitation. However, modifications of the temporal development and the local distribution of precipitation are visible and demonstrated in Fig. 11. In case of feldspar, precipitation was delayed during early cloud stages with all combinations of freezing modes. The local distribution of precipitation was not modified by additional contact freezing but significantly changed by additional deposition freezing: Precipitation was reduced in the cloud center from 75 mm to 35 mm and was spread over a wider area with 12 km diameter (instead of 4 km). This indicates the influence of deposition freezing on precipitation at the cloud edges. In contrast, for the kaolinite/Saharan dust cases, delayed precipitation from sole immersion freezing was slightly increased during later cloud stages by additional contact freezing and even more enhanced by additional contact and deposition freezing. Similarly, precipitation in the cloud center was increased from 70 mm with sole immersion freezing to 80 mm with additional contact freezing, to 95 mm with additional deposition freezing, and to 115 mm with both additional freezing modes. For the latter case Fig. 12 illustrates how additional contact and deposition modes altered the ice particle spectra. Although their impact on ice formation was rather low they modified the ice particle spectra in the way that higher numbers of larger ice particles were formed.

For the cases listed in Table 6 with lower internally mixed immersion INP fractions and higher contact and deposition INP fractions, an enhancement of total precipitation was neither found but again the development of precipitation and the local distribution were modified. In Figure 11c and 11d results from sole immersion freezing are also shown. 0.1% feldspar in the immersion mode (dashed blue line) affected a delay of precipitation while 1% additional contact and deposition INP (solid blue lines, cases 1 and 2) as well as additional biological INP (solid green line, case 3) led to an increase during early cloud stages (Fig. 11c). However, with biological INP (case 3) this increase was less significant. In contrast, the enhancement of precipitation affected by 0.1% kaolinite in the immersion mode (dashed orange line) was reduced by additional contact and deposition freezing (orange solid line, case 4) as well as by additional biological INP (solid light green line, case 5). Here the reduction was less significant with biological INP. Figure 11d again indicates that more precipitation during early cloud stages was coupled with a significant increase of precipitation in the cloud center although the total precipitation was not higher than in the reference case.

Thus, when contact and deposition freezing equally contribute to ice formation as immersion freezing (with higher INP fractions) they may work in both directions: They may enhance precipitation during early cloud stages thereby counterbalancing the delaying effect of immersion freezing. On the other hand, when immersion freezing itself shows some



"small trigger effects" these may be suppressed by contact and deposition freezing during early cloud stages. Additional biological particles counteract these tendencies.

The results presented in this section indicate that immersion freezing as the major ice forming process inhibits an increase of total precipitation above the amount from the reference case homogeneous freezing. However, additional deposition nucleation as small perturbation has the chance to modify precipitation. In dependence of the active INP types possible changes are an increase of precipitation during early cloud stages coupled with more precipitation in the cloud center or no effect on the temporal development but a spread of precipitation over a wider area beyond the cloud. Such "small trigger effects" were not confirmed for contact freezing. When contact and deposition freezing nearly equally contribute to ice formation as immersion freezing the effects do not show a clear trend. Their interactions may favorite or suppress the formation of larger ice particles and, thus, an enhancement or a delay of precipitation.

Growth processes in mixed-phase clouds are determined by the collision efficiencies of the involved drops and ice particles which in turn are dependent on the sizes of the collision partners (Pruppacher and Klett, 2010; Diehl et al., 2006). Thus, the development of drop and ice particle spectra during cloud evolution determines the effectivity of growth processes and eventually the formation of precipitation-sized hydrometeors.

## 5 Summary and conclusions

In this paper an improved version of the 3D cloud modeling system COSMO-SPECS (Grützun et al., 2008) is presented which allows to study the impact of aerosol particle types and three heterogeneous freezing modes on ice formation and precipitation. A deep convective cloud with a wide temperature range from +20°C on ground level and -50°C at cloud top was simulated. The strong vertical updraft in this cloud lifted the nucleated drops within 10 min from the zero degree level to the cloud top where the major fraction of drops froze homogeneously. In this study it was investigated if under such conditions heterogeneous freezing may have a significant impact on ice formation and how this affects precipitation. In particular, it was looked for so-called "small trigger effects", i.e. if small perturbations such as low fractions of biological particles in the immersion mode or the less efficient processes contact and deposition freezing could modify precipitation. The following conclusions were drawn:

1. The different freezing processes do hardly compete with and affect each other. Homogeneous, immersion, and contact freezing which require supercooled drops occur at different altitudes in the cloud. Deposition nucleation dominating at the highest altitudes is not in competition with homogeneous freezing because it is not coupled to supercooled drops. Also contact and deposition freezing are not in competition; they are both coupled to inactivated particles but are dominant at different altitudes. Some deposition nucleation is possible at lower heights at the cloud edges, while contact freezing concentrates rather towards the cloud center where more drops are available.

2. Precipitation is modified by the formation of larger ice particles. This is suppressed in regions where homogeneous freezing is dominant because there high numbers of small ice particles compete for growth via water vapor or drop



deposition. It may be suppressed also in median regions when the impact of immersion freezing is high, i.e. with higher fractions of efficient INP. In such cases even small drops contain already sufficient insoluble material to affect freezing. In contrast, the formation of larger ice particles by growth processes is supported in median regions of the cloud when only small fractions of immersion INP are active. Then larger drops freeze because higher masses of insoluble material in the drops are required. Additionally, supercooled drops are present for riming and the ice particles need some time to reach the melting level. On the other hand, in lower regions where contact freezing is active growth processes are hindered because of shorter times until the ice particles reach the melting level. Therefore, so-called "small trigger effects" may be initiated by immersion freezing but probably not by contact freezing.

3. Deposition nucleation affects primarily the formation of very small pristine ice particles, however, afterwards the formation of large ice particles by collision with supercooled drops is supported. In this way, deposition nucleation indirectly promotes the formation of large ice particles which modify precipitation and it may initiate "small trigger effects" as well.

4. In comparison to the reference case homogeneous freezing small perturbations may affect an enhancement of total precipitation but mostly the effects are limited to modifications of the temporal development of precipitation, i.e. a gradual increase already at early cloud stages instead a strong increase at later cloud stages. These effects are coupled with changes in the local distribution of precipitation, i.e. approximately 50% more precipitation in the cloud center. The modifications depend on the active freezing modes, the fractions of active INP, and the composition of the internal mixtures in the drops.

5. In general, precipitation from the simulated deep convective cloud did not show significant variations in the total precipitation amount. Changes in the local distribution of precipitation were more remarkable. Because of the strong vertical updraft in the present case precipitation may be determined mainly by cloud dynamics and less by the cloud microphysics.

Further simulations will switch to cloud situations with weaker cloud dynamics, i.e. reduced vertical velocity and slower ascent of the air and, finally reduced cloud top height. Thus, microphysical changes in the cloud may have more time to develop. Homogeneous freezing will have a smaller impact on ice formation while heterogeneous freezing will show higher impact.

## 6 Acknowledgements

This work is part of the research group INUIT (Ice Nuclei research UnIT) FOR1525 and was supported by the Deutsche Forschungsgemeinschaft under grant DI 1539/1-2. We appreciate the INUIT community for providing experimental data as base of parametrizations and for helpful discussions. We would like to thank Ralf Wolke and Jens Stoll from TROPOS Leipzig for providing the COSMO-SPECS code and for their support during re-starting the model and solving initial





problems. Thanks to Martin Simmel from TROPOS Leipzig for fruitful discussions and support. K. Diehl would like to thank Daniel Kunkel from IPA Mainz for his aid by installing, starting, and performing COSMO-SPECS model simulations on the high performance cluster MOGON at the University of Mainz.

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




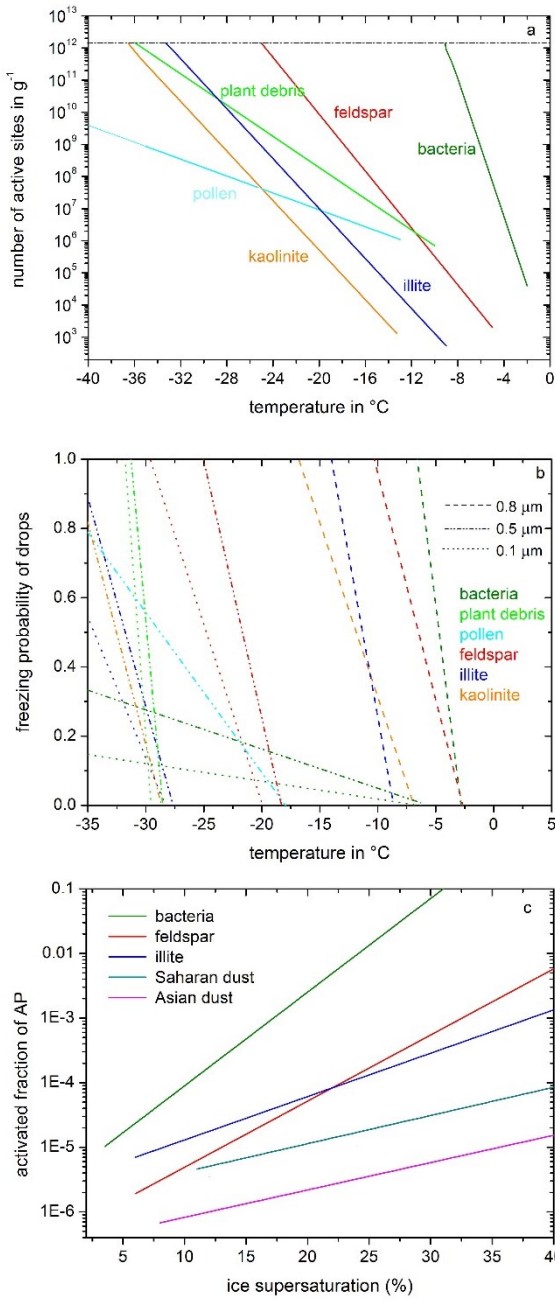

**Figure 1: Parameterizations of heterogeneous freezing. (a) Immersion freezing: numbers of active sites per unit mass as function of temperature, from Diehl and Mitra (2015), with new data for plant debris, based on Hiranuma et al. (2015). (b) Contact freezing: freezing probability of drops in the contact mode as function of temperature, from Diehl and Mitra (2015), with new data for pollen and plant debris according to Hoffmann (2015) and Hiranuma et al. (2015). (c) Deposition nucleation: activated fraction of**
10 **particles as function of ice supersaturation, from Diehl and Mitra (2015).**




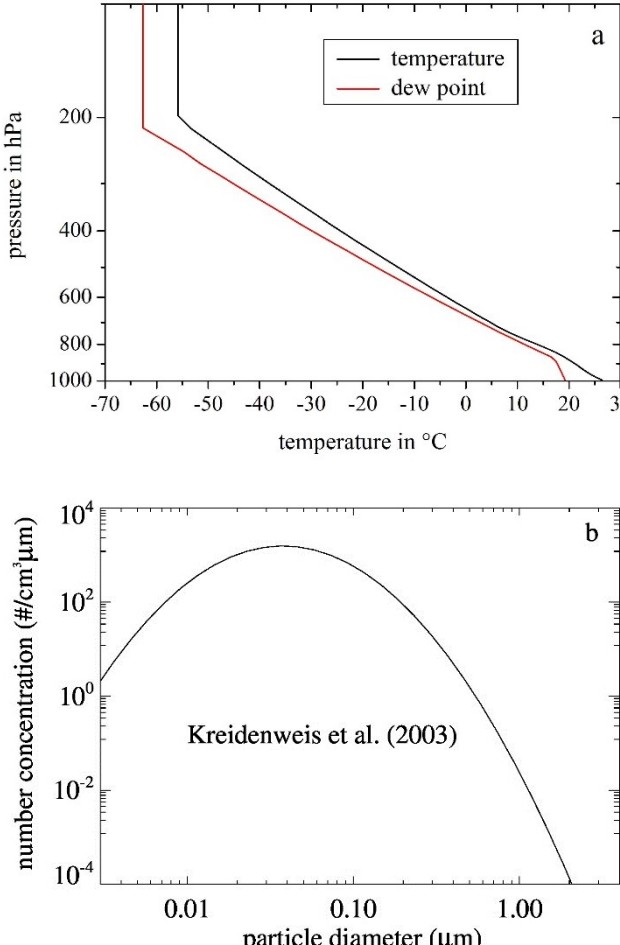

**Figure 2: Initial conditions of model simulations. (a) Vertical profiles of temperature and dew point due to Weisman and Klemp**
**(1982). (b) Initial dry particle number size distribution according to Kreidenweis et al. (2003).**



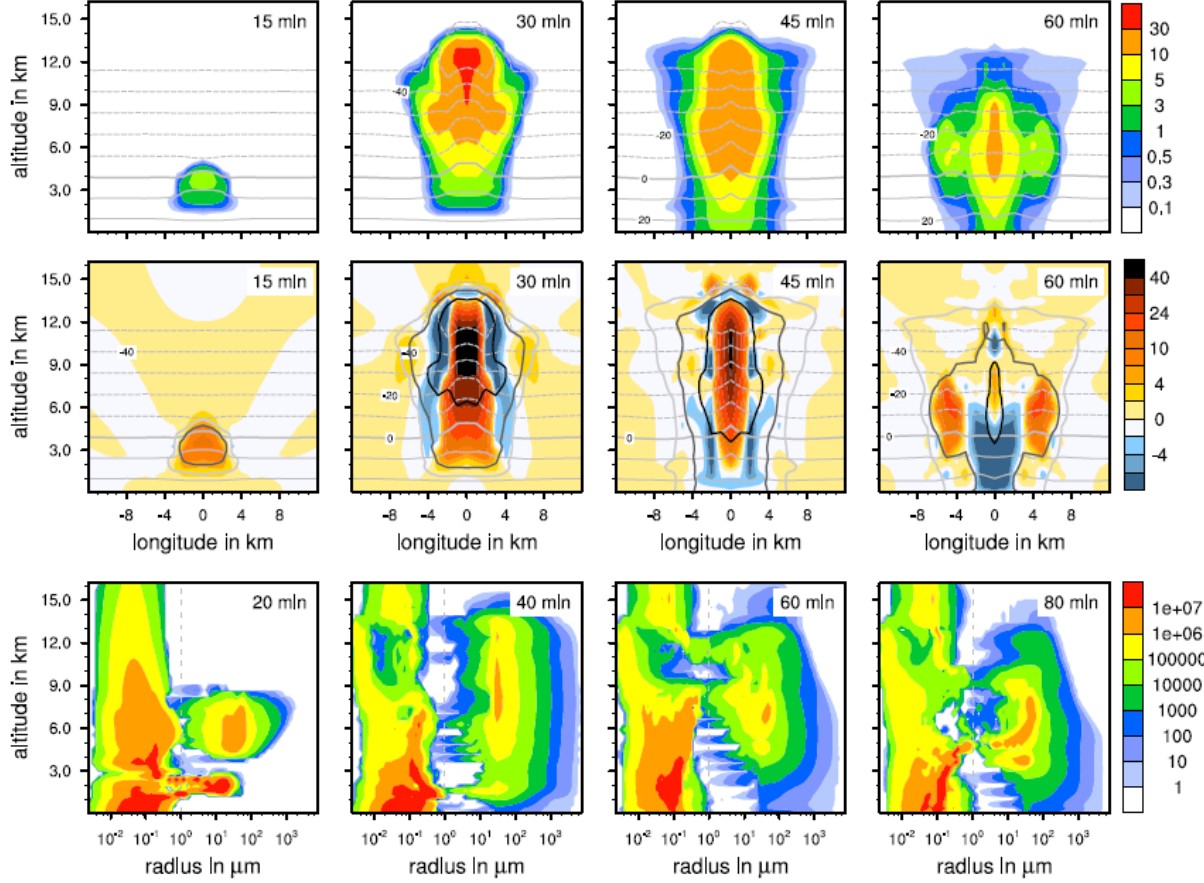

**Figure 3: Upper and middle panel: temporal development of two parameters in the warm test case shown in a vertical cut through the cloud center. Horizontal dashed lines: temperature in °C. Upper panel: liquid water content in g/kg, lower panel: vertical velocity in m/s. Solid lines in middle panel: contour lines of the total water content: 10 (black), 1 (dark grey), and 0.1 g/kg (light grey). Lower panel: temporal development of aerosol and drop size spectra in the center cell of the cloud. Number concentrations per m³.**



**Figure 4: Ice formation from the reference case homogeneous freezing. Upper and middle panel: temporal development of two parameters shown in a vertical cut through the cloud center. Horizontal dashed lines: temperature in °C. Upper panel: ice water contents in g/kg, middle panel: ice particle numbers per m³. Lower panel: ice particle size spectra in the center cell of the cloud at different times. Number concentrations per m³. Left pictures in middle and lower panel: Primary homogeneous freezing only.**





**Figure 5: Ice formation from immersion freezing with 1% feldspar. Upper and middle panel: temporal development of two parameters shown in a vertical cut through the cloud center. Horizontal dashed lines: temperature in °C. Upper panel: ice water contents in g/kg, middle panel: ice particle numbers per m³. Lower panel: ice particle size spectra in the center cell of the cloud at different times. Number concentrations per m³. Left pictures in middle and lower panel: Primary immersion freezing only.**





**Figure 6: Ice formation from contact freezing with 10% feldspar. Upper and middle panel: temporal development of two parameters shown in a vertical cut through the cloud center. Horizontal dashed lines: temperature in °C. Upper panel: ice water contents in g/kg, middle panel: ice particle numbers per m³. Lower panel: ice particle size spectra in the center cell of the cloud at different times. Number concentrations per m³. Left pictures in middle and lower panel: Primary contact freezing only.**





**Figure 7: Ice formation from deposition nucleation with 10% feldspar. Upper and middle panel: temporal development of two parameters shown in a vertical cut through the cloud center. Horizontal dashed lines: temperature in °C. Upper panel: ice water contents in g/kg, middle panel: ice particle numbers per m³. Lower panel: ice particle size spectra in the center cell of the cloud at different times. Number concentrations per m³. Left pictures in middle and lower panel: Primary deposition freezing only.**





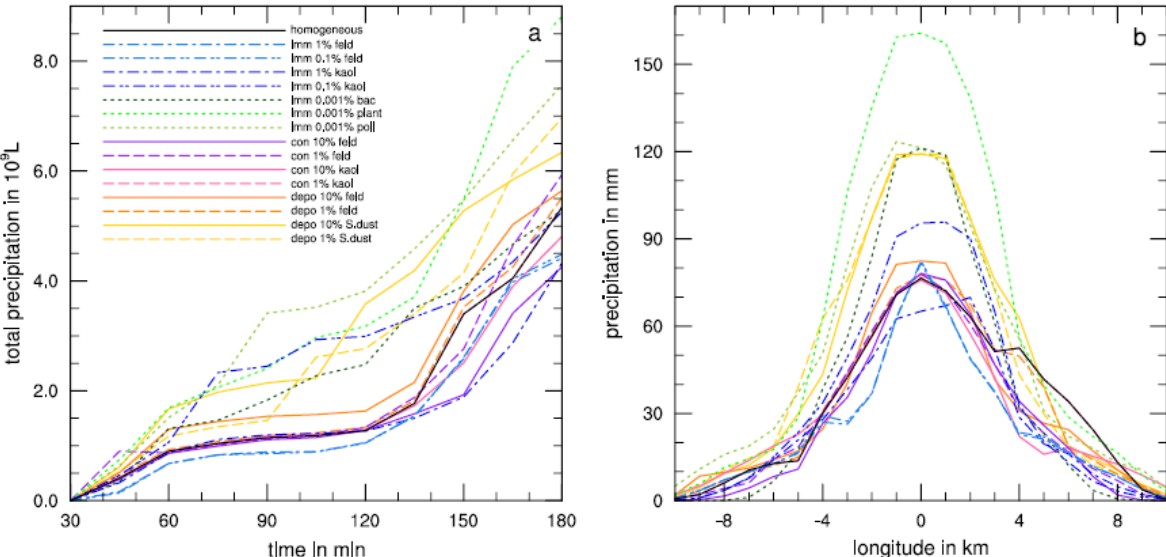

**Figure 8: (a)** Temporal development of precipitation during 180 min modeling time for coupled homogeneous and one heterogeneous freezing process. **(b)** Local distribution of precipitation through the cloud after 180 min. Black lines: reference case homogeneous freezing. Colored lines mark the coupled heterogeneous freezing mode, the line styles mark the fraction of ice-active material.





**Figure 9: Ice formation for different cases with coupled homogeneous and one heterogeneous freezing mode: case 1: immersion freezing with 1% feldspar, case 2: immersion freezing with 0.001% plant debris, case 3: contact freezing with 1% feldspar, case 4: deposition nucleation with 1% Saharan dust. Three columns on the left: ice water content in g/kg at different times shown in a vertical cut through the cloud center. Horizontal dashed lines: temperature in °C. Column on the right: ice particle size spectra in the center cell of the cloud. Number concentrations per m³.**




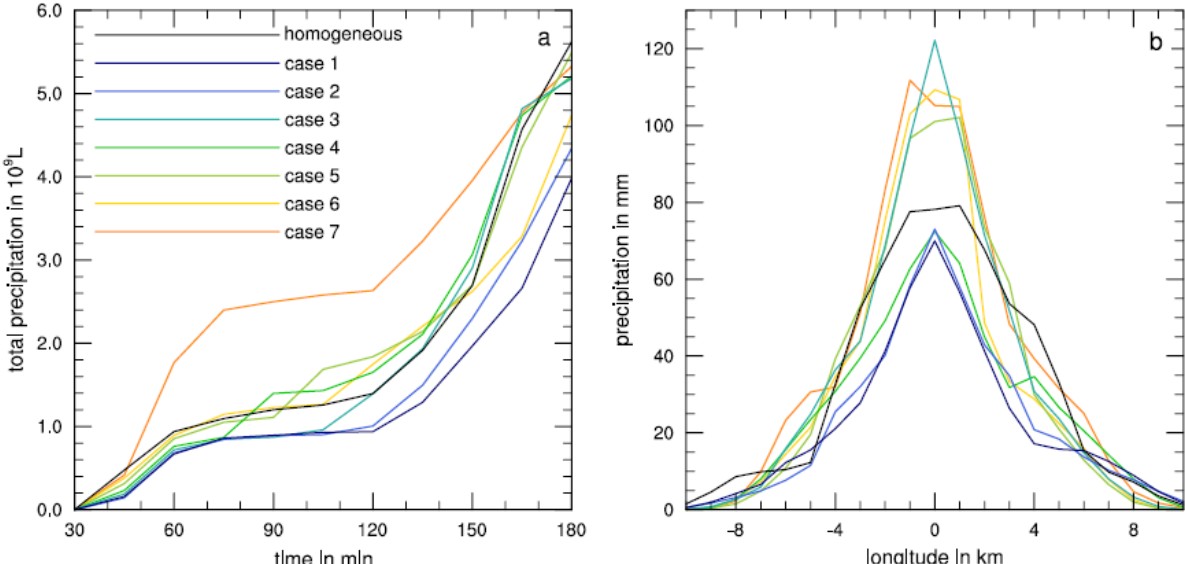

**Figure 10: (a) Temporal development of precipitation during 180 min modeling time for coupled homogeneous and internally mixed immersion freezing. (b) Local distribution of precipitation through the cloud after 180 min. Black line: reference case homogeneous freezing. Colored lines mark the different cases as listed in Table 4.**





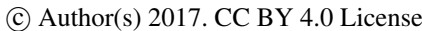

**Figure 11: Upper panel: Results from immersion freezing sole and coupled with other freezing modes. Colored lines mark the different cases as listed in Table 5. Lower panel: Results from mixed cases with coupled freezing modes. Colored solid lines mark the different cases as listed in Table 6. Dashed lines: immersion freezing only. Black lines: reference case homogeneous freezing. Left panel: Temporal development of precipitation during 180 min modeling time. Right panel: Local distribution of precipitation through the cloud after 180 min.**





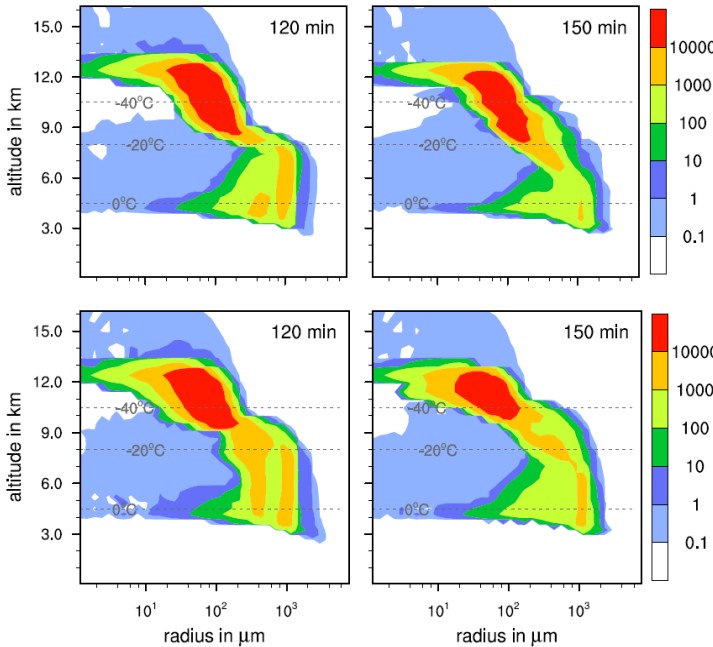

**Figure 12: Ice particle size spectra in the center cell of the cloud after 120 and 150 min for cases with 1% kaolinite/Saharan dust. Number concentrations per m³. Upper panel: sole immersion freezing, lower panel: combined immersion, deposition, and contact freezing.**



| particle type | $a_{con}$ | $b_{con}$ | $a_{con}$ | $b_{con}$ |
|---|---|---|---|---|
| **plant debris** | $0.35 < d_{ap} \leq 0.6$ µm | | $0.6$ µm $< d_{ap} \leq 1$ µm | |
| | -0.4459 | -13.176 | -0.3660 | -10.447 |
| **pollen** | − | | $0.4 < d_{ap} \leq 1$ µm | |
| | − | − | -0.0463 | -0.8305 |

**Table 1: New values of contact freezing constants in Eq. 4 for pollen and plant debris based on data from Hiranuma et al., (2015) (plant debris) and Hoffmann (2015) (pollen).**

| | immersion | contact | deposition |
|---|---|---|---|
| 10% feldspar | mixed | mixed | mixed |
| 10% kaolinite/Saharan dust | mixed | mixed | mixed |
| 1% feldspar | mixed | liquid | mixed |
| 1% kaolinite/Saharan dust | mixed | liquid | liquid |
| 0.1% feldspar | mixed | liquid | liquid |
| 0.1% kaolinite/Saharan dust | mixed | liquid | liquid |
| 0.01% bacteria | mixed | liquid | liquid |
| 0.01% plant debris | mixed | liquid | liquid |
| 0.01% pollen | liquid | liquid | liquid |
| 0.001% bacteria | mixed | liquid | liquid |
| 0.001% plant debris | liquid | liquid | liquid |
| 0.001% pollen | liquid | liquid | liquid |
| homogeneous freezing | mixed | | |

**Table 2: List of simulated cases with single freezing modes resulting in mixed-phase or liquid clouds.**




| INP type | INP fraction | total precipitation after 180 min in $10^9$ L | | |
|---|---|---|---|---|
| | | immersion | contact | deposition |
| feldspar | 10% | − | *4.26* | 5.64 |
| kaolinite/Sah. dust | 10% | − | 4.81 | 6.34 |
| feldspar | 1% | *4.41* | 5.94 | 5.52 |
| kaolinite/Sah. dust | 1% | *4.33* | 5.34 | **6.97** |
| feldspar | 0.1% | *4.48* | − | − |
| kaolinite/Sah. dust | 0.1% | 5.26 | − | − |
| bacteria | 0.001% | 5.29 | − | − |
| plant debris | 0.001% | **8.81** | − | − |
| pollen | 0.001% | **7.57** | − | − |
| homogeneous | − | 5.62 | | |

**Table 3: Total precipitation after 180 min modeling time for coupled homogeneous and one heterogeneous freezing process. Marked in bold face: cases with more than 20% enhancement of precipitation. Marked in italics: cases with more than 20% reduction of precipitation.**

| | mineral dust particles in % | | | biological particles in % | | | precipitation after 180 min × $10^9$ L |
|---|---|---|---|---|---|---|---|
| | feldspar | illite | kaolinite | bacteria | plant deb. | pollen | |
| case 1 | 1 | 1 | 3 | 0.01 | 0.01 | 0.001 | *4.00* |
| case 2 | 1 | 1 | 3 | 0.001 | 0.0001 | 0.0001 | *4.39* |
| case 3 | 0.3 | 0.3 | 0.4 | 0.001 | 0.001 | 0.001 | 5.18 |
| case 4 | 0.1 | 0.1 | 0.8 | 0.001 | 0.001 | 0.001 | 5.22 |
| case 5 | 0.01 | 0.01 | 0.9 | 0.001 | 0.001 | 0.001 | 5.50 |
| case 6 | 0.02 | 0.02 | 0.06 | 0.001 | 0.001 | 0.001 | 4.78 |
| case 7 | 0.001 | 0.001 | 0.09 | 0.001 | 0.001 | 0.001 | 5.36 |
| pure cases | 0.1 − 1 | | | − | | | 4.33 − 6.32 |
| homogeneous | − | | | | | | 5.62 |

**Table 4: Total precipitation after 180 min modeling time for coupled homogeneous and internally mixed immersion freezing. Marked in italics: cases with more than 20% reduction of precipitation.**





| freezing mode | precipitation after 180 min in $10^9$ L | |
| --- | --- | --- |
| | 1% feldspar | 1% kaolinite/Sah. dust |
| immersion | *4.41* | *4.33* |
| contact | 5.94 | 5.34 |
| deposition | 5.52 | **6.97** |
| immersion + contact | 4.98 | 4.62 |
| immersion + deposition | *4.26* | 5.02 |
| immersion + contact + deposition | *4.35* | 5.41 |
| homogeneous | 5.62 | |

**Table 5: Total precipitation after 180 min modeling time for coupled homogeneous, immersion, contact and deposition freezing in various combinations. Marked in bold face: cases with more than 20% enhancement of precipitation. Marked in italics: cases with more than 20% reduction of precipitation.**

| | freezing mode | feldspar % | illite % | kaolinite % | bacteria % | plant deb. % | pollen % | precipitation × $10^9$ L |
| --- | --- | --- | --- | --- | --- | --- | --- | --- |
| case 1 | immersion | 0.1 | 0 | 0 | 0 | 0 | 0 | 5.18 |
| | contact/depos. | 1 feldspar | | | | | | |
| case 2 | immersion | 0.09 | 0.001 | 0.001 | 0 | 0 | 0 | 5.53 |
| | contact/depos. | 1 kaolinite; 1 Saharan dust | | | 0 | | | |
| case 3 | immersion | 0.09 | 0.001 | 0.001 | 0.001 | 0.001 | 0.001 | *3.62* |
| | contact/depos. | 0.1 plant debris; 0.1 bacteria | | | | | | |
| case 4 | immersion | 0.001 | 0.001 | 0.09 | 0 | 0 | 0 | *4.21* |
| | contact/depos. | 1 kaolinite; 1 Saharan dust | | | | | | |
| case 5 | immersion | 0.001 | 0.001 | 0.09 | 0.001 | 0.001 | 0.001 | 5.34 |
| | contact/depos. | 0.1 plant debris; 0.1 bacteria | | | | | | |
| homogeneous | | − | | | | | | 5.62 |

**Table 6: Total precipitation after 180 min modeling time for mixed cases with coupled homogeneous, immersion, contact and deposition freezing. Marked in italics: cases with more than 20% reduction of precipitation.**