# Peer review of "Model simulations with COSMO-SPECS: Impact of heterogeneous freezing modes and ice nucleating particle types on ice formation and precipitation in a deep convective cloud."

_Atmospheric Chemistry and Physics, 2017_

## Referee Comment (RC1) · Anonymous Referee #2 · 2 Aug 2017

The manuscript presents model simulations of deep convective clouds, using the 3D cloud resolving model COSMO-SPECS with an improved parameterization of ice forming processes. The effects of small changes in different heterogeneous freezing modes on ice formation, cloud microphysical properties and precipitation are investigated, the latter with respect to the time of occurrence, spatial distribution and total amount. In general the manuscript provided by Diehl and Grützun aims the scope of ACP and is scientifically relevant for the atmospheric science community.

The manuscript is well structured and I had hardly any issued with the approach and

style of presentation. I appreciate the extensive experiments conducted and the detailed presentation of the results, and I have only some minor questions and comments.

Minor comments:

- The effects of latent heat release due to phase change are not discussed. As stated in Grützun et al., 2008 this may become important for the development of the cloud, with respect to updraft velocities. Are there significant difference between the different model runs used in this study?

- Would your findings be applicable in terms of cloud seeding of e.g. severe thunderstorms by the reduction of total precipitation amount and/or more equal temporal and spatial distribution of precipitation?

- Abstract, page 1, line 23 – 24: Be more specific here, e.g. mention that an enhancement/reduction of more than 20% in total precipitation amount was found for different freezing modes/particle types.

- Page 2, line 15 – 16: How large is the fraction of homogeneously formed ice? Could you add an example with a reference (e.g. the reference from page 8, line 4 – 5).

- Page 2, line 27: Are "the most important INP types" referring exclusively to deep convective cloud? If so, please specify here. Up to date it is not solved which are the "most important INP type" for all cloud types, e.g. the importance of biological INP in the free troposphere is questioned, and marine aerosols might have a global contribution in remote areas.

- Page 6, line 5 – 6: In the presented work it is not considered that biological ice-nucleating active macromolecules (INMs) as small as 10 nm can be released

from their carriers, e.g. from pollen and fungal spores, upon contact with water, and can be released in much higher concentrations. Thus they can have a much higher atmospheric implication as previously assumed. Would an increased biological/pollen concentration influence your results, resp. would you expect a larger effect of biological particles, and which effect would this have on the importance of different freezing modes and could this finally influence precipitation modification?

- Page 11, line 3: Is the term "similar" justified here, since an enhancement/reduction of 20% was observed?

- Page 15, line 19 – 22: I find this statement confusing, since earlier on it is mentioned that deviations in the total precipitation amount was $\pm$ 20%. Did you perform statistical tests to come to the conclusion that this is not a significant difference? Also you are concluding that "precipitation may be determined mainly by cloud dynamics", as an outcome of the updraft velocities up to 40 $ms^{-1}$. This is somewhat contradicting your chosen model setup to study effects on precipitation by cloud microphysics. To be coherent I would rather conclude that, although you have the strong dynamical component, cloud microphysics still can influence precipitation.

---

## Referee Comment (RC2) · Anonymous Referee #1 · 8 Aug 2017

The present manuscript investigates the effect of various heterogeneous freezing/nucleation modes for primary ice production in an idealized convective cloud case with 3D simulations using COSMO-SPECS with a horizontal resolution of 1 km. The model benefits from the bin-microphysics (SPECS) scheme, which allows an accurate representation of microphysics and the distinction between the different freezing modes. Sensitivity simulations with warm-microphysics only, homogeneous freezing solely, single heterogeneous freezing modes (immersion freezing, condensation freezing, deposition nucleation and contact freezing) with various aerosol concentrations and afterwards combined with homogeneous freezing, were conducted. The single freezing modes are most prominent under different conditions (temperature and present liquid cloud droplets) and thus do not directly compete with each other. The sensitivity of precipitation formation on single heterogeneous freezing modes combined with homogeneous freezing was analysed with various aerosol concentrations. The additional heterogeneous freezing modes can increase and decrease the precipitation amount compared to homogeneous freezing only. The onset time of precipitation and the accumulated precipitation rate is affected less strong than location distribution and intensity. The concept of testing different freezing modes sequentially in a bin-microphyisics scheme is done in an accurate way and therefore this article qualifies for publication in ACP.

**General comments**

Do your results agree or disagree with the current literature? e.g Hande 2016 testing also different freezing modes for several cloud types including convective cases. ... I think the present study could benefit from comparison to other studies.

Also, better curve out the advantage that the bin-microphysics has compared to often used 2-moment microphysics.

P2; L20-21:

*small trigger effects* are mentioned frequently in the course of this article. Can you give a detailed explanation what these small trigger effects are? Is it the triggered glaciation process of the cloud, which includes ice growth, Wegener-Bergeron-Findeisen process, multiplication processes, collision beside others? Can you give an idea on how sensitive those triggered processes are to parameters that need to be set e.g. for collision processes, in the model? How sensitive are model results to the setup of this parameters compared to the sensitivity on the here tested heterogeneous freezing processes? Is the feedback on dynamics also one of the triggered effects?

P3; L16:

*The original COSMO model works with a Kessler-type ...*    That is true, but for scientific purpose on investigating cloud microphysics more sophisticated microphysical schemes are used, such as two-moment schemes for warm and cold microphysics. Can you recommend on the advantage of your bin-microphysics approach compared to often used two-moment schemes?

Motivate why heterogeneous freezing modes are important although homogeneous freezing is dominant in convective clouds.

P8; L10-28:

The definition from Korolev 2003 distinguishes between liquid, mixed and ice cloud. I do not agree with the definition of mixed-phase clouds with IWF above 0.1. That also includes completely glaciated clouds. How do you define the mixing ratio? Is that the whole water column, so the ice water path and liquid water path, when you say integrated ice water content? Is that comparable to the definition from Korolev 2003, which refeers to the mixing ratio within the observational volume? See also the updated article about mixed-phase cloud definition Korolev et al 2017.

What does this analysis tell? Does contact freezing hardly nucleate any ice crystals? Are the few INP directly sedimenting out?

If have the feeling the purpose of this analysis using IWF is to figure out if the single freezing modes can produce a sufficient amount of ice. That could be included in **Ice water contents**.

**Specific comments**

P1; L10:

delete: as essential process

P1; L22-27:
Sentence to complicate: Try something like: In comparison to the reference case, with homogeneous freezing only, such small perturbations due to additional heterogeneous freezing rather effect the total precipitation amount. The temporal development and local distribution/ structure of precipitation are more likely affected by such perturbations.

P1; L22:
collision with pristine ice particles; How are secondary ice processes considered in the model?

P3; L18-20: Is wind transported? Is transportation (advection) of temperature and pressure the only process considered in the dynamical core? Which timestep was exactly used in the present study (10s or 100s)?

P2: L21-22:
Again, which timestep is used?

P8;L24 - P9;L2; P11;L24-L27:
Please avoid description of the Figure, which should be part of the caption in the text. Give interpretation instead.

Figure 3:
The description is confusing. Do you mean the middle panel with: Lower panel: vertical velocities . . . ?

[Figure]

Figure 8:
The text in the legend is not focussed, hard to read and there are too many information on this plot. Can you make two plots; e.g one for immersion mode and another for contact and deposition?

P11; L11: Is it: In some cases **the accumulated** precipitation **amount** stayed constant during the next hour and increased at later times. So that mean **no precipitation**?

P11; L21-25: Why do you call the simulations now cases 1-4 without intuitive names. Can you analyse the cloud properties with the same simulations as done for precipitation, to be consistent? As far as I understand some of the *cases* are the same as used before for precipitation analysis. E.g. imm 0.001% plant is case 2 now?

Figure 11:
I think this plot should show how additional freezing modes contribute to the amount, location and temporal evolution of precipitation. This is done for two aerosol setups. Make a clear distinction between the two sets e.g by line pattern or even in two separated plots.

Table 3:
Think about showing the deviations from the reference simulations. Would be easier to catch which modes contribute to enhancement of precipitation amount or suppresses precip.

**Technical corrections**

P1; L10: In deep convective clouds, heavy rain **is** often formed involving the ice phase.

P2; L29: acronym BAP not necessary because never used again

P3; L28: fraction, $\epsilon$

P4; L26: temperature, $T$.

P4; L27: delete temperature

Eq: 1: Insert that $n_\mathrm{m}$ is a function of $T$

P5; L3: delete: () in citation Diehl and Mitra

Equations in general: Some look blurred, in general derivations, d, should not be italic as well as subscripts.

P6; L18; $\epsilon$ already introduced, delete soluble fraction

P6; L19: INP (replace ice nucleating particles with INP everywhere after INP was introduced. Check also all other abbreviations.)

P10 and following: You could use ice crystal number concentration (ICNC) instead of description like numbers of ice particles per m$^3$. eg. ICNC up to $10^{-4}\,\mathrm{m}^{-3}$ were reached ...

P11;L30: 1a ; delete 1

P12;L16: Internally mixed INP in the immersion mode; Is this another subsection? I guess numbers are missing.
* * *

---

## Author Comment (AC1) · 14 Dec 2017

Dear Editor,

we would like to thank you for handling the review of our manuscript. We would also like to thank the reviewers for their suggestions which help to improve the new version of the manuscript.

Please find below a list of both reviewers' comments followed by our responses and any manuscript changes.

**Responses to Reviewer 1**

General comments:

*Do your results agree or disagree with the current literature? e.g. Hande 2016 testing also different freezing modes for several cloud types including convective cases. ... I think the present study could benefit from comparison to other studies.*

Due to the reviewer's suggestion, comparisons with other model simulations have been included in the revised paper, i.e. results from Hiron and Flossmann (2015), Hande et al. (2107), and Hande and Hoose (2017). These are:

Section 4.2.2:

"This correlation between relative humidity and contact freezing effects was found also in the model studies of Hande et al. (2017). "

Section 4.2.3:

"Model simulations performed by Hande and Hoose (2017) for convective clouds resulted in similar findings: immersion freezing was the most important heterogeneous freezing mode. In regions with temperatures below -35°C homogeneous freezing was dominant while there were less effects from deposition freezing. Contact freezing was important at warmer temperatures."

Section 4.3.1:

"Similar observations were made by Hiron and Flossmann (2015) who studied the role of heterogeneous freezing modes in the framework of a 1.5D bin-resolved cloud model. They simulated a convective cloud which reached an altitude of 9.5 km with temperatures below -40°C. In single cases with sole contact and deposition freezing they found more accumulated rain as with sole immersion and homogeneous freezing."

Section 5 (Conclusions)

"5. The results indicate so-called "small trigger effects" of heterogeneous freezing in comparison to homogeneous freezing as well as "small trigger effects" of deposition and contact freezing in comparison to immersion freezing. Therefore, although homogeneous freezing is dominant in a deep convective cloud, heterogeneous freezing processes should not be neglected. Besides immersion freezing contact and deposition freezing are also important. This finding is in contrast to the conclusion of Hiron and Flossmann (2015). From the fact that contact and deposition freezing contributed only low amounts of ice particles in comparison to the other freezing modes they concluded that they might be neglected in cloud models with less complexity.

"Small trigger effects" of biological particles in comparison to mineral dust particles were not found, thus, the role of biological particles in atmospheric clouds remains still unclear. This confirms the conclusion of Hiron and Flossmann (2015) regarding the role of biological particles. They studied a case with sole bacterial INP acting in a non-specific ice nucleation mode which resulted in significantly higher amounts of precipitation as all other cases. However, when bacteria were added in a simulation where all other INP were also forming ice their influence became negligible."

*Also, better curve out the advantage that the bin-microphysics has compared to often used 2-moment microphysics.*

Please see comment below, regarding P3.

*P2; L20-21: small trigger effects are mentioned frequently in the course of this article. Can you give a detailed explanation what these small trigger effects are? Is it the triggered glaciation process of the cloud, which includes ice growth, Wegener-Bergeron-Findeisen process, multiplication processes, collision beside others? Can you give an idea on how sensitive those triggered processes are to parameters that need to be set e.g. for collision processes, in the model? How sensitive are model results to the setup of this parameters compared to the sensitivity on the here tested heterogeneous freezing processes?*

Yes, small trigger effects mean triggered glaciation of the cloud, including the Bergeron-Findeisen process and growth by collision processes. Ice multiplication is not included in the present model version of COSMO-SPECS.

Certainly these effects are influenced by the description of collision processes in the model. As mentioned in the paper, collision processes are described by a linear discrete method including collision kernels with collision efficiencies based on tabulated experimental data combined with theoretical calculations (Simmel et al., 2002; Kerkweg et al., 2003). Corresponding densities and terminal velocities are used regarding liquid drops, ice particles, and aerosol particles. So these parameters affect the growth processes of ice particles in the model. However, we would estimate that the model results are more sensitive to the different heterogeneous freezing modes, the ice nucleating particle types, and the ice-active fraction of the particles than to parameters describing collision processes.

A new paragraph has been added to the revised paper to explain what it meant by small trigger effects:

"This situation may create so-called "small trigger effects", i.e. small perturbations which do not show significant effects on the first sight may trigger cloud microphysical responses. For instance, a small number of ice particles is formed by a small amount of ice nucleating particles. They grow further by the deposition of water vapor, including the effects of the Bergeron-Findeisen process, i.e. at the expense of liquid drops. With increasing sizes of the ice particles collisions with supercooled drops become more likely (Pruppacher and Klett, 2010). The ice particles grow by riming when they collide with smaller supercooled drops which are deposited on the ice surface and subsequently freeze. When small ice particles collide with larger supercooled drops, the latter freeze by contact-induced nucleation of the ice particle. In this way even small amounts of ice particles may efficiently modify the distribution of ice and liquid water in a cloud. Thus, even when homogeneous freezing is dominant in deep convective clouds, additional heterogeneous freezing, in particular taking place in lower cloud regions, may have essential impact on ice formation."

*Is the feedback on dynamics also one of the triggered effects?*

Possible feedback of small trigger effects on the cloud dynamics were investigated but not found. A comparison of the vertical velocities for various cases indicated that there are no significant differences. The major difference is between the warm case without freezing and the case with homogeneous freezing: the field with highest vertical velocity of 40 m/s is vertically extended in the homogeneous case in comparison to the warm case, i.e. ice formation effects additional updraft in higher altitudes. In cases with different ice formation there are only small local changes. Therefore, this topic was not discussed in the paper. In the revised manuscript, a short explanation is included in the conclusions.

"2. Regarding the vertical velocity in the cloud, because of the release of latent heat during freezing the fields of highest vertical updraft were vertically extended in comparison to the warm case (see

Fig. 3) by the additional release of latent heat during freezing. However, cases with different ice formation resulted in small local changes only. "

*P3; L16: The original COSMO model works with a Kessler-type ... That is true, but for scientific purpose on investigating cloud microphysics more sophisticated microphysical schemes are used, such as two-moment schemes for warm and cold microphysics. Can you recommend on the advantage of your bin-microphysics approach compared to often used two-moment schemes?*

A new paragraph has been added to the revised manuscript describing two-moment and spectral bin schemes in more detail including the advantage of bin schemes for the present simulations.

"The original COSMO model works with a Kessler-type cloud microphysics bulk scheme. This includes various states of water such as cloud and rain water, several forms of ice but takes into account mass densities only (Kessler, 1995). Later two–moment schemes were developed which additionally consider the hydrometeor number concentrations (e.g., Seifert and Beheng, 2006). Those schemes predict the evolution of mass as well as number densities of several hydrometeor types. However, they offer only limited potential to include the aerosol particles. In spectral bin schemes the particle mass is discretized so that the hydrometeor spectra are divided into size bins where number and mass are considered (e.g., Reisin et al., 1996; Khain et al., 2004; Simmel and Wurzler, 2006). In those schemes initial aerosol particle spectra are explicitly included and the particle and drop/ice particle spectra evolve freely. Thus, spectral microphysical schemes allow detailed investigations of aerosol-cloud interactions. In particular, when the ice phase is included, explicit information about drop and ice particle sizes and the development of size spectra are given allowing conclusions about the correlations of ice formation and precipitation."

*Motivate why heterogeneous freezing modes are important although homogeneous freezing is dominant in convective clouds.*

Please see comment above about small trigger effects and the new paragraph in the revised paper.

*P8; L10-28: The definition from Korolev 2003 distinguishes between liquid, mixed and ice cloud. I do not agree with the definition of mixed-phase clouds with IWF above 0.1. That also includes completely glaciated clouds.*

The reviewer is right, the explanation of ice clouds was missing in the paper. The mentioned sentence has been changed:

"A liquid cloud is defined by $IWF < 0.1$, a mixed-phase cloud by $0.1 \leq IWF \leq 0.9$, and an ice cloud by $IWF > 0.9$."

*How do you define the mixing ratio? Is that the whole water column, so the ice water path and liquid water path, when you say integrated ice water content? Is that comparable to the definition from Korolev 2003, which refers to the mixing ratio within the observational volume? See also the updated article about mixed-phase cloud definition Korolev et al 2017. What does this analysis tell? Does contact freezing hardly nucleate any ice crystals? Are the few INP directly sedimenting out? If have the feeling the purpose of this analysis using IWF is to figure out if the single freezing modes can produce a sufficient amount of ice. That could be included in ice water contents.*

To calculate the ice water fraction, not the liquid and ice water paths were used but the liquid and ice water contents. Writing the term "integrated" was possibly misleading and has been removed. The purpose of this analysis was to show how efficient the different freezing modes are in combination with various fractions of ice-active material. It indicated that contact and deposition freezing are much less efficient than immersion freezing and require higher fractions of ice-active material. Based on this analysis it was decided which cases were suited for studying small trigger

effects: such where a liquid cloud formed but one step higher (i.e. one magnitude higher fractions of INP) mixed phase clouds were formed. It would have been superfluous to study, e.g., 0.1% kaolinite in the contact mode, when even 1% kaolinite affects a liquid cloud only. In particular for biological particles, still 0.1% bacteria in the contact mode would not affect a mixed phase cloud, therefore, investigating cases with realistic fractions of 0.001% biological particles in the contact mode seemed unnecessary.

Yes, the analysis shows that contact as well as deposition freezing hardly nucleate any ice particles. One reason is that both freezing modes require interstitial aerosol particles which are always present in the simulated cloud because of entrainment but their numbers are low. Therefore, contact and deposition freezing take place preferably in regions where the relative humidity is low, e.g., at the cloud edges. This is discussed in the paper in Section 4.2.2. Of course the ice water content indicates if the single freezing modes produce sufficient amounts of ice, however, the ice water fraction gives more insight which cases are reasonable to be studied further.

Specific comments

*P1; L10: delete: as essential process*

This wording has been deleted.

*P1; L22-27: Sentence to complicate: Try something like: In comparison to the reference case, with homogeneous freezing only, such small perturbations due to additional heterogeneous freezing rather effect the total precipitation amount. The temporal development and local distribution/ structure of precipitation are more likely affected by such perturbations.*

The mentioned sentence has been changed into:

"In comparison to the reference case, with homogeneous freezing only, such small perturbations due to additional heterogeneous freezing rather effect the total precipitation amount. More likely the temporal development and the local distribution of precipitation are affected by such perturbations. This results in a gradual increase of precipitation already at early cloud stages instead of a strong increase at later cloud stages, coupled with approximately 50% more precipitation in the cloud center."

*P1; L22: collision with pristine ice particles; How are secondary ice processes considered in the model?*

We are not sure how to answer this question. Collision processes in the model are described by the linear discrete method of Simmel et al. (2002) and include the collision kernel of Kerkweg et al. (2003). For collisions between ice particles and drops, the corresponding densities and terminal velocities are used. If during the model run an ice particle collides with a supercooled drop, instantaneous freezing of the involved drop is assumed. Their masses are added and the new ice particle is sorted into the corresponding size bin of the ice particle spectrum while the previous ice particle and drop are removed from their bins.

*P3; L18-20: Is wind transported? Is transportation (advection) of temperature and pressure the only process considered in the dynamical core?*

The winds u, v, and w as well as temperature and pressure are transported. This part is completely computed within the COSMO framework using the original advection routines.

"Within the COSMO model the horizontal and vertical winds as well as temperature and pressure are transported within a time step of 1 to 100 s leading to dynamically updated values."

*Which timestep was exactly used in the present study (10s or 100s)?*

*P2: L21-22: Again, which timestep is used?*

The time range 10 to 100 s was a mistake, it should be 1 to 100 s. This has been corrected. The time steps actually used in the present model studies have been added in the revised manuscript:

"… are transported within a time step of 1 to 100 s leading to dynamically updated values. These are used for the microphysical loop which consists of time steps of 1 s or smaller where changes in the hydrometeor spectra due to the included microphysical processes are calculated (Grützun et al., 2008). In the present simulations, the dynamical and the microphysical time steps were 4 s and 1 s, respectively, i.e. within one dynamical time step four microphysical time steps were calculated."

*P8;L24 - P9;L2; P11;L24-L27: Please avoid description of the Figure, which should be part of the caption in the text. Give interpretation instead.*

The mentioned paragraphs were changed according to the reviewer's suggestion; unnecessary descriptions of the figures have been removed from the revised version.

"Figures 4 to 7 show the corresponding results of ice formation: ice water contents, ice particle numbers, and ice particle size spectra."

"To illustrate how ice formation influences the total condensed water in the deep convective cloud and, thus, precipitation, results from four example cases shown in Figure 8 are considered in more detail."

"Figure 9 shows results from the cases 1 to 4 in each panel: ice water contents and ice particle numbers."

*Figure 3: The description is confusing. Do you mean the middle panel with: Lower panel: vertical velocities . . . ?*

The reviewer is right, this was a mistake. The vertical velocity is shown in the middle panel. This has been corrected in the figure caption.

*Figure 8: The text in the legend is not focussed, hard to read and there are too many information on this plot. Can you make two plots; e.g one for immersion mode and another for contact and deposition?*

We decided to keep all three modes together in one plot; this allows a better comparison. The differences would not be easily visible in separate plots. The legend of the final figure will be improved to be better readable.

*P11; L11: Is it: In some cases the accumulated precipitation amount stayed constant during the next hour and increased at later times. So that mean no precipitation?*

It was meant that precipitation stayed almost constant, i.e. increased only slightly. This has been corrected.

*P11; L21-25: Why do you call the simulations now cases 1-4 without intuitive names. Can you analyse the cloud properties with the same simulations as done for precipitation, to be consistent? As far as I understand some of the cases are the same as used before for precipitation analysis. E.g. imm 0.001% plant is case 2 now?*

Cases 1 to 4 are cases from Table 3 and Figure 8. The case numbers were introduced so that the explanations are better understandable. The mentioned paragraph has been partly re-written:

"To illustrate how ice formation influences the total condensed water in the deep convective cloud and, thus, precipitation, results from four example cases shown in Figure 8 are considered in more detail. These are immersion with 1% feldspar (case 1), immersion with 0.001% plant debris (case 2), contact with 1% feldspar (case 3), and deposition with 1% Saharan dust (case 4). The amounts of total precipitation were $4.33 \times 10^9$ L, $8.81 \times 10^9$ L, $5.34 \times 10^9$ L, and $6.97 \times 10^9$ L, respectively (see Table 3)."

*Figure 11: I think this plot should show how additional freezing modes contribute to the amount, location and temporal evolution of precipitation. This is done for two aerosol setups. Make a clear distinction between the two sets e.g by line pattern or even in two separated plots.*

In the final figure, the two aerosol sets will be presented by more distinct colors and line patterns.

*Table 3: Think about showing the deviations from the reference simulations. Would be easier to catch which modes contribute to enhancement of precipitation amount or suppresses precip.*

We do not exactly understand what the reviewer means. Cases with enhancement of precipitation are written in bold face, e.g. there is at least 20% more precipitation than in the reference case. Cases with suppression of precipitation are shown in italics, e.g. there is at least 20% less precipitation than in the reference case.

Technical corrections

Most technical corrections as suggested by the reviewer have been included in the revised paper; if not explanations are given below.

*P1; L10: In deep convective clouds, heavy rain is often formed involving the ice phase.*

*P2; L29: acronym BAP not necessary because never used again*

*P3; L28: fraction, _*

*P4; L26: temperature, T.*

*P4; L27: delete temperature*

*Eq. 1: Insert that nm is a function of T*

*P5; L3: delete: () in citation Diehl and Mitra*

*Equations in general: Some look blurred, in general derivations, d, should not be italic as well as subscripts.*

*P6; L18; _ already introduced, delete soluble fraction*

*P6; L19: INP (replace ice nucleating particles with INP everywhere after INP was introduced. Check also all other abbreviations.)*

We prefer to use abbreviations not everywhere during the text, only when an expression is repeated several times.

*P10 and following: You could use ice crystal number concentration (ICNC) instead of description like numbers of ice particles per m3. e.g. ICNC up to 10–4 m–3 were reached ...*

The expression ice crystals would not be correct as there are no crystal shapes included in the model description.

*P11;L30: 1a ; delete 1*

Unfortunately, we did not find 1a on the mentioned page.

*P12;L16: Internally mixed INP in the immersion mode; is this another subsection? I guess numbers are missing.*

It is another subsection, but according to ACP guidelines only three levels of sectioning are allowed.

**Responses to Reviewer 2**

Minor comments:

*The effects of latent heat release due to phase change are not discussed. As stated in Grützun et al., 2008 this may become important for the development of the cloud, with respect to updraft velocities. Are there significant difference between the different model runs used in this study?*

A comparison of the vertical velocities for various cases indicated that there are no significant differences. Due to cases with different ice formation there are only small local changes observed: the field with highest vertical velocity of 40 m/s is somewhat extended in the homogeneous case in comparison to the warm case, and in the coupled homogeneous + immersion case in comparison to the homogeneous case. Therefore, this topic was not discussed in the paper. In the revised manuscript, a short explanation is included in the conclusions.

"2. Regarding the vertical velocity in the cloud, because of the release of latent heat during freezing the fields of highest vertical updraft were vertically extended in comparison to the warm case (see Fig. 3) by the additional release of latent heat during freezing. However, cases with different ice formation resulted in small local changes only. "

*Would your findings be applicable in terms of cloud seeding of e.g. severe thunderstorms by the reduction of total precipitation amount and/or more equal temporal and spatial distribution of precipitation?*

We are no experts in cloud seeding and, therefore, aware of recommending any applications of the results from our sensitivity studies. However, it might be possible to suppress heavy precipitation by adding high amount of effective INP to a thunderstorm cloud as this might reduce the formation of large ice particles.

*Abstract, page 1, line 23 – 24: Be more specific here, e.g. mention that an enhancement/reduction of more than 20% in total precipitation amount was found for different freezing modes/particle types.*

A significant enhancement of total precipitation was observed for some cases with single heterogeneous freezing only which do not really represent atmospheric clouds. Therefore, we decided to leave the Abstract as it is. See also our reply to your last comment.

*Page 2, line 15 – 16: How large is the fraction of homogeneously formed ice? Could you add an example with a reference (e.g. the reference from page 8, line 4 – 5).*

We could not find a reference where a defined number fraction of homogeneous freezing is given, even in the mentioned reference (Phillips et al., 2007). It is always mentioned that homogeneous freezing is dominant. How large the actual fraction is depends surely on a number of parameters, therefore, it would be somewhat inadequately to give a clear value.

*Page 2, line 27: Are "the most important INP types" referring exclusively to deep convective cloud? If so, please specify here. Up to date it is not solved which are the "most important INP type" for all cloud types, e.g. the importance of biological INP in the free troposphere is questioned, and marine aerosols might have a global contribution in remote areas.*

The "most important INP types" refer to atmospheric clouds in general. This is stated more clearly in the revised manuscript.

"The most important atmospheric INP types are mineral dust and biological particles …"

*Page 6, line 5 – 6: In the presented work it is not considered that biological ice-nucleating active macromolecules (INMs) as small as 10 nm can be released from their carriers, e.g. from pollen and fungal spores, upon contact with water, and can be released in much higher concentrations. Thus they can have a much higher atmospheric implication as previously assumed. Would an increased biological/pollen concentration influence your results, resp. would you expect a larger effect of biological particles, and which effect would this have on the importance of different freezing modes and could this finally influence precipitation modification?*

The used parameterization of immersion freezing is based on the numbers of active sites per mass contained in the drops. Here the particle size was not limited but just the particle mass per drop was considered. As shown in Table 2, in the immersion mode biological particles with ice-active fractions as low as 0.001% (bacteria) or 0.01% (plant debris) had an essential impact on ice formation. Assuming higher values of $F_{INP}$ in the model simulations as currently used, i.e. more than 0.001%, would certainly lead to similar effects as higher fractions of mineral dust particles, i.e. a delayed increase of precipitation during early cloud stages and reduced total precipitation.

For contact and deposition freezing, the sizes of possible biological INP were limited to be not smaller than 0.3 µm (see end of Section 2.2.3). If INP with sizes down to 10 nm would be assumed as part of the initial aerosol particle distribution this would increase the numbers of possible INP by a factor of 2 at least (compare the Kreidenweis particle spectrum in Figure 2). As shown in Table 2, even with some somewhat higher biological fractions of 0.01% only liquid clouds were formed with contact and deposition modes. This picture would certainly change if the lower size limit of the INP was set to 10 nm in the way that the biological particles could have large effects on ice formation via contact and deposition modes. But this depends still on the fraction of interstitial dry particles which are required for contact and deposition freezing. How this could finally effect precipitation is not easy to conclude from the present simulations; this requires new sensitivity studies. Another question is if the presently used parameterizations of contact and deposition freezing are valid for such small INP also. However, this reviewer's comment gives a valuable motivation for future simulations.

*Page 11, line 3: Is the term "similar" justified here, since an enhancement/reduction of 20% was observed?*

We assume that this was a misunderstanding. The results summarized in Table 3 indicate that **in most cases the total precipitation amount was similar** to homogeneous freezing but there were **some cases with more than 20% deviations** in both directions.

*Page 15, line 19 – 22: I find this statement confusing, since earlier on it is mentioned that deviations in the total precipitation amount was _ 20%. Did you perform statistical tests to come to the conclusion that this is not a significant difference? Also you are concluding that "precipitation may be determined mainly by cloud dynamics", as an outcome of the updraft velocities up to 40 m/s. This is somewhat contradicting your chosen model setup to study effects on precipitation by cloud microphysics. To be coherent I would rather conclude that, although you have the strong dynamical component, cloud microphysics still can influence precipitation.*

What we meant was that significant enhancement of total precipitation was observed for some cases with single heterogeneous freezing only which do not really represent atmospheric clouds. In cases with combined homogenous + heterogeneous freezing no enhancement of total precipitation was observed but modifications of the temporal development and the local distribution. The mentioned section in the conclusion was re-written according to the reviewer's suggestion.

"In general, precipitation from the simulated deep convective cloud did not show significant variations in the total precipitation amount. Changes in the local distribution of precipitation were more remarkable. Because of the strong vertical updraft in the present case precipitation is strongly influenced by cloud dynamics but cloud microphysics still has an important impact."